# Transcription Factor Homeobox D9 Drives the Malignant Phenotype of HPV18-Positive Cervical Cancer Cells via Binding to the Viral Early Promoter

**DOI:** 10.3390/cancers13184613

**Published:** 2021-09-15

**Authors:** Shigenori Hayashi, Takashi Iwata, Ryotaro Imagawa, Masaki Sugawara, Guanliang Chen, Satoko Tanimoto, Yo Sugawara, Ikumo Tanaka, Tomoya Matsui, Hiroshi Nishio, Masaru Nakamura, Yuki Katoh, Seiichiro Mori, Iwao Kukimoto, Daisuke Aoki

**Affiliations:** 1Department of Obstetrics and Gynecology, Keio University School of Medicine, 35 Shinanomachi, Shinjuku-ku, Tokyo 160-8582, Japan; shigenorih1974@yahoo.co.jp (S.H.); imagawa0322@keio.jp (R.I.); masaki.sugawara@keio.jp (M.S.); guanliangc@gmail.com (G.C.); m0105052@yahoo.co.jp (S.T.); wanko.vs.nyanko@ezweb.ne.jp (Y.S.); ikumo_tanaka@yahoo.co.jp (I.T.); tomoya.nc.us@gmail.com (T.M.); nishio@z3.keio.jp (H.N.); nakamura127533045@hotmail.com (M.N.); yukikatoh@keio.jp (Y.K.); aoki@z7.keio.jp (D.A.); 2Department of Functional Morphology, Nihon University School of Medicine, 30-1 Oyaguchikamicho, Itabashi-ku, Tokyo 174-8610, Japan; 3Pathogen Genomics Center, National Institute of Infectious Diseases, 4-7-1, Gakuen, Musashimurayama 208-0011, Japan; moris@nih.go.jp (S.M.); ikuki@nih.go.jp (I.K.)

**Keywords:** cervical cancer, HOXD9, HPV18, *P105* promoter

## Abstract

**Simple Summary:**

Transcription factor homeobox D9 (HOXD9) was previously reported to bind to the *P97* promoter of HPV16 to induce viral *E6*/*E7* oncogenes. In this article, we investigated whether HOXD9 regulated the *P105* promoter of HPV18 and examined the role of HOXD9 in intracellular signaling of cervical cancer (CC). HOXD9 was directly bound to the *P105* promoter and regulated the expression of *E6*/*E7* genes of HPV18. The HOXD9 knockdown suppressed the *E6*/*E7* gene expression in HPV18-positive cervical cancer cells. It decreased the expression of *E6*, activated the p53 pathway, and induced apoptosis. In addition, downregulation of the *E7* gene expression activated the Rb pathway, causing G1 arrest in the cell cycle and markedly suppressing cell proliferation. Our results indicate that HOXD9 has pivotal roles in the proliferation and immortalization of HPV18-positive cervical cancer cells through activating the *P105* promoter.

**Abstract:**

Persistent infections with two types of human papillomaviruses (HPV), HPV16 and HPV18, are the most common cause of cervical cancer (CC). Two viral early genes, *E6* and *E7*, are associated with tumor development, and expressions of *E6* and *E7* are primarily regulated by a single viral promoter: *P97* in HPV16 and *P105* in HPV18. We previously demonstrated that the homeobox D9 (HOXD9) transcription factor is responsible for the malignancy of HPV16-positive CC cell lines via binding to the *P97* promoter. Here, we investigated whether HOXD9 is also involved in the regulation of the *P105* promoter using two HPV18-positive CC cell lines, SKG-I and HeLa. Following the HOXD9 knockdown, cell viability was significantly reduced, and *E6* expression was suppressed and was accompanied by increased protein levels of P53, while mRNA levels of *TP53* did not change. *E7* expression was also downregulated and, while mRNA levels of *RB1* and *E2F* were unchanged, mRNA levels of E2F-target genes, *MCM2* and *PCNA*, were decreased, which indicates that the HOXD9 knockdown downregulates *E7* expression, thus leading to an inactivation of E2F and the cell-cycle arrest. Chromatin immunoprecipitation and promoter reporter assays confirmed that HOXD9 is directly associated with the *P105* promoter. Collectively, our results reveal that HOXD9 drives the HPV18 early promoter activity to promote proliferation and immortalization of the CC cells.

## 1. Introduction

Cervical cancer (CC) is the fourth most frequent cancer in women worldwide regarding both incidence and mortality, with 569,847 women diagnosed with CC and 311,365 deaths in 2018 [1]. Human papillomavirus (HPV) is detected in more than 90% of CC cases and is considered the most important cause of cervical carcinogenesis. There are approximately 40 types of HPV that infect the female genital organs, 20 types of which, HPV16, 18, 26, 31, 33, 35, 39, 45, 51, 52, 56, 58, 59, 66, 67, 68, 70, 73 and 82, are considered carcinogenic and classified as high-risk types [2]. HPV16 is the most frequently detected type in CC, accounting for approximately 50% of cases, followed by HPV18, accounting for 20% [3].

HPV is a small, double-stranded DNA virus that has a genome of approximately 8000 bp. The viral genome consists of early genes (*E1*, *E2*, *E4*, *E5*, *E6*, and *E7*) that are involved in viral gene expression, and late genes (*L1* and *L2*) that encode the capsid proteins. The *E6* and *E7* genes are particularly involved in tumorigenesis. The E6 protein promotes the degradation of the tumor suppressor P53, and the E7 protein inactivates the RB tumor suppressor, thereby inducing cell immortalization and genetic abnormalities [4]. The E6 protein promotes cell proliferation by stimulating degradation of P53 via formation of a trimeric complex of E6, P53, and the cellular ubiquitination enzyme E6-AP [5,6]. The E6-induced degradation of P53 interferes with the biological functions of P53, thereby disrupting the control of cell cycle progression and leading to increased tumor cell proliferation [7]. RB protein binds to the E2F growth factor and suppresses its transcriptional activity. The E7 protein binds to and degrades RB, suppresses an association between RB and E2F, and activates E2F, leading to deregulated cell proliferation and immortalization [7,8].

Suppression of *E6* and *E7* gene expression in CC cells results in the arrest of cell proliferation, apoptosis, and cell death. [9]. Therefore, if specific drugs targeting *E6* and *E7* can be developed, it is expected that these drugs can be applied to the treatment of CC; however, to date, no such drugs that represses *E6* and *E7* have been developed. In HPVs, the expression of all early genes, including *E6* and *E7*, is regulated by a single early promoter, which has sequence variations according to the HPV type. Therefore, repression of this early promoter can suppress the expression of all early genes. This mechanism is unique to the HPV virus, and suppressors that target the early promoter are supposed to be an excellent molecular target therapy for humans without side effects [10].

The HOX gene family includes transcription factors with highly conserved gene sequences of 183 bp, which encode nuclear proteins called homeoproteins. To date, 39 HOX genes have been identified and classified into clusters of HOXA, HOXB, HOXC, and HOXD according to the chromosomal locus. In recent years, HOX genes have attracted attention in the field of cancer, and it has been reported that they are overexpressed in many cancers and are closely related to cancer growth, invasion, metastasis, and anticancer drug resistance [11,12]. *HOXD9* is a HOXD gene located on 2q31. This gene participates in the development and patterning of the forelimb and axial skeleton [13]. HOXD9 enhances hepatocellular carcinoma (HCC) cell migration, invasion, and metastasis via the *ZEB1* gene expression [14]. Moreover, HOXD9 promotes epithelial–mesenchymal transition of HCC and colorectal cancer cells [11,12]. Another study reported that *HOXD9* is involved in glioma cell proliferation and survival and is highly expressed in a side population of cancer stem-like cells [15]. We previously reported that HOXD9 is a factor involved in the malignant traits of CC and its molecular mechanism in a HPV type 16-positive CC. *HOXD9* knockout in HPV16-positive CC cell lines inhibited the activity of the *P97* promoter, the early HPV promoter of HPV16, simultaneously suppressing the expression of *E6* and *E7*, resulting in the arrest of cell proliferation and the induction of apoptosis [16]. In this study, we investigated the role of *HOXD9* in survival, proliferation, and metastasis of HPV18-positive CC, and its effect on HPV18 oncogene promoter *P105*. Cervical adenocarcinoma, small cell carcinoma, and neuroendocrine carcinoma, which are considered to have worse prognoses than squamous cell carcinoma, are mostly caused by HPV types 16 and 18, and other high-risk HPVs are extremely rare [17,18]. This research is expected to lead to the development of new therapies, not only for squamous cell carcinoma, but also for these intractable histological types.

## 2. Methods and Methods

### 2.1. Cell Lines

The SKG-I CC cell line was established by our group and deposited in the cell bank of the National Institutes of Biomedical Innovation (Japan). HeLa cells were purchased from the American Type Culture Collection (ATCC, Manassas, VA, USA) and maintained in our laboratory. These two cell lines are both HPV18-positive. To confirm the identity of the analyzed cell lines, we performed short terminal repeat (STR) genotyping that revealed a correspondence of >80% of the tested markers. SKG-I cells were cultured in Ham’s F-12 medium (Sigma-Aldrich, St. Louis, MI, USA) supplemented with 10% FBS and 1% penicillin–streptomycin. HeLa cells were cultured in Dulbecco’s modified Eagle’s medium (DMEM; Nacalai Tesque, Kyoto, Japan) supplemented with 10% FBS and 1% penicillin–streptomycin.

### 2.2. Knockdown of HOXD9

pLKO.1 vector incorporating including short hairpin RNA (shRNA) targeting human *HOXD9* gene (NM_014213.3) with the following sequence was obtained from Sigma–Aldrich: 5′-CCGGGCAGCAACTTGACCCAAACAACTCGAGTTGTTTGGGTCAAGTTGCTGCTTTTT-3′. Empty pLKO.1 was used as a negative control. The shRNA and two packaging plasmids, pCMV-VSV-G-Rev (Addgene, Watertown, MA, USA) and pMDL-g/p-RRE (Addgene), were transfected into 293T cells. HilyMax (Dojindo, Kumamoto, Japan) was used to transfect the constructs. The supernatant was collected at 48 h post-transfection and concentrated by polyethylene glycol precipitation using a Lenti-XTM Concentrator (Clontec Laboratories, Mountain View, CA, USA). The recombinant lentivirus-containing medium was added to HeLa and SKG-I cells.

### 2.3. RT-PCR and qPCR

Total RNA was extracted from CC cell lines using the RNeasy Mini Kit (Qiagen, Venlo, Netherlands). Quantitative reverse transcription PCR (RT-PCR) was performed according to the standard protocol. *GAPDH* (Hs.PT.39a.22214836) expression was used to normalize qPCR results. TaqMan RT-PCR primers and probes for human *HOXD9* (Hs00610725_g1), HPV18 E6/7, and *RB1* (Hs00153108_m1) were purchased from Applied Biosystems (Waltham, MA, USA). Human *TP53* (Hs.PT.58.39676686), *E2F1* (Hs.PT.58.45513742), *MCM2* (Hs.PT.58.46729588), and *PCNA* (Hs.PT.58.24924035) probes were purchased from Integrated DNA Technologies (Coralville, IA, USA).

### 2.4. Cell Viability Assay

Cell viability was assessed by the water-soluble tetrazolium salt (WST-1) cytotoxic assay (Roche Diagnostics, Tokyo, Japan). Treated 1.0 × 10^4^ HeLa and SKG-I cells were incubated in a medium containing WST-1 at the working concentration stated in the user manual. After 4 h, the change in optical absorbance was recorded by a microplate reader at 450 nm with 650 nm as the reference wavelength.

### 2.5. Cell Cycle Analysis by Flow Cytometry

Cell cycle analysis in HeLa and SKG-I cells was performed by flow cytometry using Vybrant DyeCycle Violet Stain (Invitrogen, San Diego, MA, USA) according to the manufacturer’s instructions. Cytographs were analyzed using Kaluza (Beckman Coulter, Brea, CA, USA).

### 2.6. Apoptosis Assay

Apoptosis analysis was performed using Annexin V-fluorescein isothiocyanate (FITC) and 7-aminoactinomycin D (7-AAD) (BD Pharmingen, Franklin Lakes, NJ, USA). To detect phosphatidylserine externalization, 5 × 10^5^ HeLa and SKG-I cells at 120 h after lentivirus infection were harvested by trypsinization, washed with PBS, and resuspended in 500 µL binding buffer (100 mM HEPES (pH7.4), 140 mM NaCl, and 2.5 mM CaCl_2_). After 15 min of Annexin V incubation at room temperature in the dark, 7-AAD was added to the samples. Finally, the samples were analyzed by the Gallios flow cytometer (Beckman Coulter).

### 2.7. Western Blotting

Western blotting was performed according to the standard protocol. The following antibodies were used: anti-GAPDH antibody (Santa Cruz Biotechnology, Santa Cruz, TX, USA) and anti-p53 antibody (Santa Cruz Biotechnology). Proteins were detected by incubation with 1:1000 dilutions of primary antibodies and anti-mouse HRP antibodies (Thermo Fisher Scientific, Waltham, MA, USA).

### 2.8. Luciferase Reporter Assay

The *P105* reporter plasmid pGL3-P105 was obtained from the Pathogen Genomics Center, National Institute of Infectious Diseases (Tokyo, Japan). Transfected HeLa and SKG-I cells (1 × 10^4^) were seeded into 96-well plates. The cells were transfected with 40 ng luciferase reporter plasmid using FuGENE-6 reagent (Promega, Madison, WI, USA). To monitor the transfection efficiency, the cells were cotransfected with 5 ng herpes simplex virus thymidine kinase promoter-driven *Renilla*-luciferase plasmid (Promega). Firefly and *Renilla* luciferase activities were measured at 48 h after transfection using the Dual-Glo Luciferase Assay Kit (Promega) and Envision Multilabel Reader (PerkinElmer Science, Waltham, MA, USA).

### 2.9. Chromatin Immunoprecipitation

Chromatin immunoprecipitation (ChIP) assays were conducted using the EZ ChIP kit (Upstate Biotechnology, Lake Placid, NY, USA) according to the manufacture’s protocol. HeLa and SKG-I cells (3 × 10^5^) were cultured for 24 h and then transfected with 8 μg pCMV-shHoxD9-c-Myc-DDK (OriGene Technologies, Rockville, MD, USA) using FuGENE-6. Samples were subjected to immunoprecipitation using mouse anti-c-Myc or normal mouse IgG (Santa Cruz Biotechnology) and analyzed by 35-cycle PCR. The PCR primers of the *P105* promoter were as follows: forward 5′-AAGGGAGTGACCGAAAACG-3′; reverse 5′-AAAGCGCGCCATAGTATTGT-3′.

### 2.10. DNA Microarray and Ingenuity Pathway Analysis

Gene expression in mock and shHOXD9 SKG-I cells was analyzed by microarray (Appendix A), and the source data were available in the GEO repository, accession no. GSE183553. For DNA microarray analysis, 0.5 µg total RNA was amplified and labeled using an Amino Allyl MessageAmp™ IIa RNA Amplification kit (Applied Biosystems). Each sample of aRNA labeled with cyanine (Cy)3 and reference aRNA labeled with Cy5 were cohybridized with a 3D-Gene™ Human Oligo chip 25k (Toray Industries Inc., Tokyo, Japan) at 37 °C for 16 h. After hybridization, each DNA chip was washed and dried. Hybridization signals derived from Cy3 and Cy5 were scanned using Scan Array Express (PerkinElmer,). The gene expression profiles were examined by analyzing scanned image using GenePixR Pro (MDS Analytical Technologies, Sunnyvale, CA, USA). All analyzed data were scaled by global normalization. Based on changes in the expression of all genes, the top 20 activated upstream regulators were predicted by Ingenuity Pathway Analysis (IPA; Ingenuity Systems; http://www.ingenuity.com, accessed on 21 April 2021.

### 2.11. Statistical Analysis

All statistical analyses were performed using GraphPad Prism 9 software (GraphPad Software, San Diego, CA, USA). Comparisons between the two groups were assessed using unpaired or paired (for matched comparisons) two-tailed Student’s *t*-tests or non-parametric Mann–Whitney U-tests. Data are presented as mean ± standard error. *p* < 0.05 was considered statistically significant.

## 3. Results

### 3.1. HOXD9 Is Essential for Proliferation of CC cells

In vitro assays were performed using SKG-I and HeLa cell lines that are positive for HPV18 with high expression levels of the *HOXD9* gene. The effects of *HOXD9* on the proliferation of CC cells in vitro was evaluated using WST-1. Cell proliferation was markedly suppressed in *HOXD9*-knockdown cells (Figure 1A,B), and WST-1 assay revealed that cell viability was significantly decreased (Figure 1C).

### 3.2. HOXD9 Represses E6 Gene Expression and Activates the p53 Pathway

The HPV18 *E6* gene was investigated by qPCR in *HOXD9*-knockdown CC cells. HPV18 *E6* gene expression was markedly suppressed in SKG-I and HeLa cells upon inhibition of *HOXD9* (Figure 2A,B).

In CC, the p53 protein is degraded by the HPV E6 protein via E6AP, and apoptosis is inhibited [4]. We examined the expression of *p53* and P53 protein after the suppression of *HOXD9* expression. When *HOXD9* expression was inhibited, *p53* mRNA expression was not significantly different between control and *HOXD9*-knockdown cells (Figure 2C). In contrast, P53 protein expression was enhanced in SKG-I and HeLa cells (Figure 2D and Appendix A). On the basis of these results, *HOXD9* was not considered to suppress the expression of the *p53* gene but was involved in the degradation of the P53 protein through upregulation of E6. Apoptosis detection by annexin V/7-AAD double staining was performed to investigate apoptosis in SKG-I and HeLa cells following *HOXD9* knockdown. The proportion of cells in the early apoptotic stage (annexin V+/7-AAD-) was increased from 2.0% to 6.5% among SKG-I cells and from 3.0% to 3.5% among HeLa cells. The proportion of cells in the late apoptotic stage (annexin V+/7-AAD+) was also increased from 3.4% to 16.0% among SKG-I cells and from 7.7% to 19.2% among HeLa cells (Figure 2E). These results suggested that knockdown of *HOXD9* expression induced apoptosis in CC cells by activating the p53 pathway. We investigated the cellular and molecular mechanisms of HOXD9 in CC cells by performing molecular and cellular functional analyses using the IPA program. To investigate the possible biological interactions of differentially regulated genes, datasets representing genes with altered expression profiles derived from microarray analysis were imported into the IPA tool. Table 1 lists the top 20 activated upstream regulators identified by IPA. In the IPA analysis, 1136 pathways were analyzed and p53 was the fifth most upregulated pathway among all pathways.

### 3.3. HOXD9 Represses E7 Gene Expression and Activates the E2F Pathway

In CC, HPV E7 protein binds to RB protein, inactivates RB, and activates the transcription factor E2F, a G1 gatekeeper [4]. If HOXD9 activates the *P105* promoter to induce the expression of the *E7* gene, then knockout of *HOXD9* will suppress *E7* gene expression and restore the tumor suppressor function of RB. This should result in the inhibition of the E2F target gene.

When HOXD9 expression was downregulated, HPV18 *E7* gene expression was significantly suppressed (Figure 3A), but *RB1* and *E2F1* gene expression was not changed (Figure 3B,C). In contrast, *MCM2* and *PCNA* gene expression were both markedly suppressed (Figure 3D,E). These genes are the targets of E2F and promote the transition of the cell cycle from the G1 to S phase [19,20,21]. We next performed cell cycle analyses. In both SKG-I and HeLa cells, the proportion of cells in the G1 phase increased, whereas cells in the S phase decreased in *HOXD9*-knockdown cells compared with the control cells (Figure 3F). In the IPA analysis, the RB pathway was selected as the 12th pathway to be activated by *HOXD9* knockout (Table 1), while E2F was selected as the 12th pathway to be inhibited (Appendix A).

### 3.4. HOXD9 Regulates the P105 Promoter by Direct Binding

We examined whether HOXD9 directly regulates the *P105* promoter. A reporter assay using the pGL3-P105 reporter plasmid was performed. When pGL3-P105 was transfected into control and *HOXD9*-knockdown SKG-I and HeLa cells, luciferase activity was significantly decreased in *HOXD9*-knockdown cells compared with control cells (Figure 4A). These results indicate that HOXD9 is an enhancer of the *P105* promoter.

We next performed a ChIP assay to investigate whether HOXD9 binds directly to the *P105* promoter. Since there was no HOXD9 antibody available for the chip assay, a c-Myc-labeled HOXD9 expression plasmid (pCMV-HOXD9-c-Myc-DDK) was transfected into SKG-I and HeLa cells to express the c-Myc-tagged HOXD9 protein. The ChIP assay was performed using the anti-c-Myc antibody. The *P105* promoter was detected in the immunoprecipitation product of both cell lines (Figure 4B and Appendix A). These results indicated that HOXD9 directly binds to the *P105* promoter region of HPV18 and promotes the expression of *E6*/*E7* genes.

## 4. Discussion

HPV-associated carcinogenesis is closely linked to the expression of the viral E6 and E7 oncoproteins [22]. The failure of cellular regulatory mechanisms involved in the control of HPV oncogene transcription and in the concomitant increase in E6 and E7 expression is believed to have a critical role in the process of HPV-associated carcinogenesis [23]. The expression of both E6 and E7 is regulated by the early promoter. The regulation of the early promoter can either be positive through the binding of transcription factors, such as AP-1 [24], progesterone [25], TEF-1 [26,27], Skn-1a [7], and NF1 [28], or negative by binding of the CDP/cut protein [29] and the YY1 protein [30]. To our knowledge, HOXD9 is the only newly identified factor regulating the early promoter since the report of CDP/Cut in 2000. In our previous study, we reported that HOXD9 promotes the expression of E6 and E7 by regulating the *P97* promoter, the early promoter of HPV16, and is involved in the growth, invasion, and metastasis of CC. In this study, we showed that HOXD9 regulated the early promoter of HPV18 as well as HPV16 in CC, and HOXD9 suppression caused cell death.

The E6 protein binds to E6-AP, which degrades the P53 protein and suppresses p53 signaling [4]. In this study, we first examined the effect of HOXD9 downregulation on p53 signaling in HPV18-positive CC cell lines. HOXD9 knockdown decreased the expression of E6 but did not change the mRNA levels of the *p53* gene, while increasing the levels of the P53 protein. This indicates that the HOXD9 knockdown inhibits E6 expression, which prevents the degradation of P53 and stabilizes the P53 protein. The apoptosis assay by flow cytometry showed an increase in the number of early apoptotic cells (Annexin ^high^ and 7AAD^low^) and dead cells (Annexin^high^ and 7AAD^high^). This suggests that the HOXD9 knockdown restores P53 protein expression and induces apoptosis. Changes in overall gene expression in HOXD9-inhibited cell lines were comprehensively analyzed by DNA microarray. Using IPA to analyze candidate regulatory genes with a high probability of involvement, the p53 pathway was selected as the fifth most upregulated pathway among all intracellular signals. This result suggests that HOXD9 downregulation suppressed *E6* gene expression, activated the p53 pathway, and induced apoptosis.

We then examined the effect of HOXD9 downregulation of E7 on RB signaling. In the cell cycle, E2F activation is required for the transition from the G1 phase to the S phase. In normal cells, the RB protein binds to E2F and inactivates it, thereby suppressing the transition from G1 to the S phase. In the presence of E7 in CC, RB binds to E7 rather than E2F. As a result, E2F is separated from RB and becomes the active form. Because the target genes of E2F are those involved in the G1 checkpoint, such as *MCM2*, *PCNA*, *hTERT*, and *CCNE1*, activation of E2F causes cells to enter S phase and promote mitosis [19,21,31,32]. In the HOXD9 knockout cell line, the expression of *E7* was decreased, but the expression of *RB1* and *E2F* was not changed, and the expression of *MCM2* and *PCNA*, the target genes of E2F, was decreased. When pathway analysis was performed by IPA in the HOXD9 knockdown cell line, the RB pathway was selected as the 12th most activated pathway and E2F was selected as the 12th most inactivated pathway (Table 1 and Appendix A). This result again confirms that HOXD9 suppression reduces E7 expression, restoring the ability of the RB protein to bind and inactivate E2F. Cell cycle studies showed that HOXD9 suppression increased the number of cells in the G1 phase. This may be due to a decrease in the expression of genes encoding growth factors involved in the G1 checkpoint, which are target genes of E2F, resulting in G1 arrest.

In summary, suppression of HOXD9 in cervical cancer can simultaneously suppress the HPV oncogenes *E6* and *E7*, arrest cell proliferation, and induce cell death. Moreover, it has been recently pointed out that E6/E7 oncoproteins of high-risk HPV can upregulate the programmed cell death-1/programmed cell death-tligand 1 (PD-1/PD-L1) axis [33]. Upregulated levels of PD-L1 in cervical cancer are associated with poorer disease-free and disease-specific survival rates in comparison to those with normal or low PD-L1 levels [34,35]. If the expression of *E6*/*E7* is suppressed by HOXD9 inhibition, the expression of PD-L1 in CC may be reduced and the cytotoxicity of T cells may be enhanced. Thus, if HOXD9 inhibitors can be clinically applied, they are expected to exert their anti-tumor effects through various pathways.

Further studies are needed to clarify the association between HOXD9 and the carcinogenesis of CC. In this study, we confirmed that HOXD9 directly binds to and activates the *P105* promoter by promoter and ChIP assays. The HOX gene cluster is known to recognize and bind to the TAAT sequence of target genes [36]. The *P105* promoter also contains TAAT sequences, and it is possible that HOXD9 binds to this site, but further studies are required to confirm this. Furthermore, the HOX family genes are known to act as transcription factors by forming complexes with other proteins [11]. Previously, we transfected 293T cells, a cell line that does not express HOXD9, with the *HOXD9* expression plasmid and the *P97* reporter plasmid pGL3-P97, and performed a promoter assay, but the luciferase activity was not enhanced [16]. Therefore, to elucidate the mechanism by which HOXD9 regulates early promoters, it is necessary to identify not only the promoter binding sites but also the cofactors required for HOXD9 function. Second, it is necessary to examine whether HOXD9 universally regulates the early genes of not only HPV type 16 and 18 but also other HPV types. If HOXD9 regulates the early promoters of many HPV types, it will be possible to develop therapeutic agents for all HPV-caused diseases. The third is a problem of using HOXD9 inhibitors for treatment. HOXD9 is a transcription factor and is thought to regulate many genes. Systemic administration of inhibitors may cause changes in the expression of several human genes as well as HPV genes, resulting in unexpected side effects. Careful studies in animals are needed to determine the effects of HOXD9 inhibitors on the human body.

## 5. Conclusions

It is known that the majority of CCs are positive for high-risk HPVs that involve two oncoproteins, E6, and E7, which regulate cell-cycle and tumor-suppressor genes, thereby affecting apoptosis and cell death. In this study, we found that HOXD9 has a critical role in the carcinogenesis of HPV18-positive CC, as with HPV16-positive CC, which we previously reported. In CC, various HPV types are responsible for squamous cell carcinoma, and the ratio of HPV types 16 and 18 is approximately 70%. In adenocarcinoma, small cell carcinoma, and neuroendocrine carcinoma, only two HPV types, 16 and 18, are detected, and other high-risk HPV types are very rarely detected. HOXD9 is a promising candidate for molecular targeted therapy for CC, including refractory histological types.

## Figures and Tables

**Figure 1 cancers-13-04613-f001:**
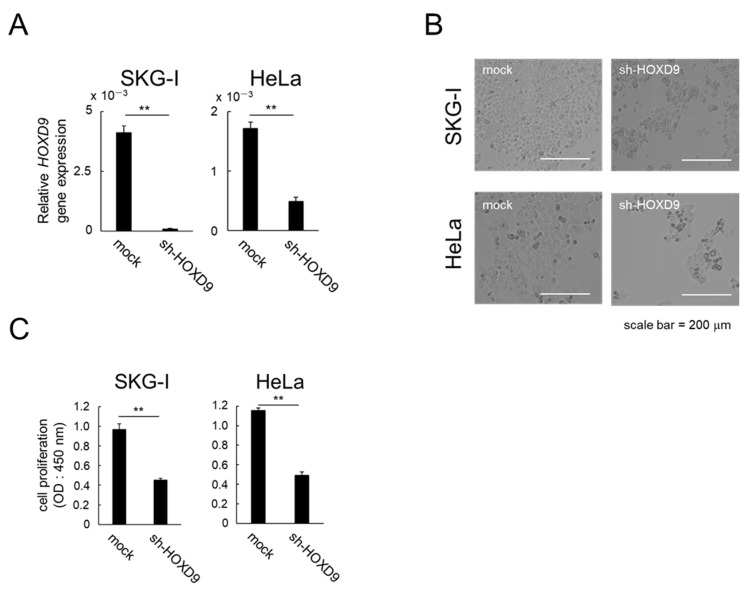
HOXD9 regulates cell proliferation in CC cell lines. SKG-I and HeLa cells were transfected with Scheme 9. (sh-HOXD9) or shRNA-empty vector (mock). (**A**) Total RNA was extracted and *HOXD9* mRNA was evaluated by real-time RT-PCR at 48 h post-transfection. (means ± SD; *n* = 3)(**B**,**C**) Proliferation of SKG-I and HeLa cells measured by microscopic observation (**B**) and WST-1 assay (**C**) (means ± SD; *n* = 3). ** *p* < 0.01.

**Figure 2 cancers-13-04613-f002:**
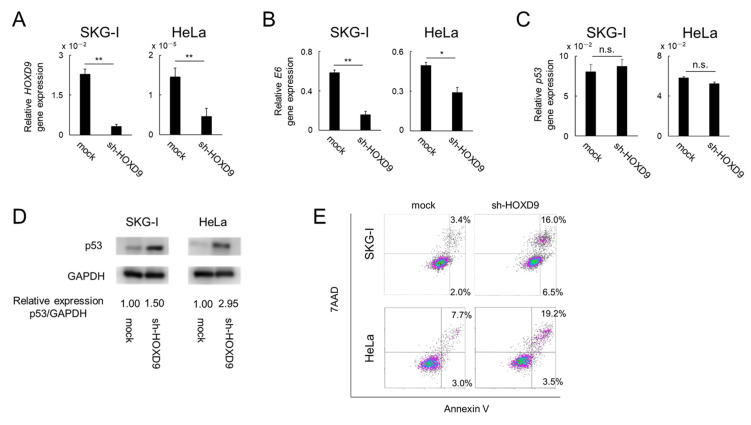
Knockdown of *HOXD9* gene induces apoptosis through low E6 expression and suppression of p53 degradation. SKG-I and HeLa cells were transfected with shRNA-HOXD9 (sh-HOXD9) or shRNA-empty vector (mock). (**A**–**C**) Total RNA was extracted and *HOXD9* (**A**), *E6* (**B**), and *TP53* (**C**) mRNA was evaluated by real-time RT-PCR at 48 h post-transfection. (means ± SD; *n* = 3) (**D**) Extracted proteins were separated by SDS-PAGE and transferred to PVDF membranes. The blotted proteins were probed with anti-GAPDH and anti-p53 antibodies. (**E**) Apoptotic monitoring was performed by staining with FITC-conjugated annexin V and 7-AAD. The percentages of annexin V+/7-AAD- cells (representing cells in the early stage of apoptosis) and annexin V+/7-AAD+ cells (representing cells in the late stage of apoptosis) were determined by flow cytometry. * *p* < 0.05, ** *p* < 0.01. n.s.: not significant.

**Figure 3 cancers-13-04613-f003:**
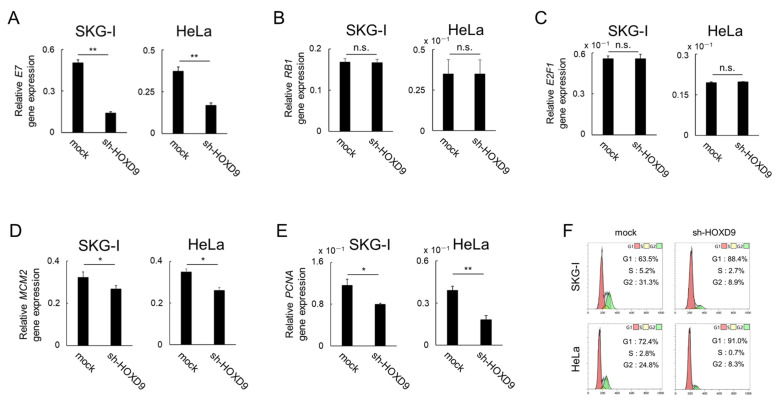
Knockdown of *HOXD9* gene causes G1 arrest through low E7 expression and decreased E2F activity. SKG-I and HeLa cells were transfected with shRNA-HOXD9 (sh-HOXD9) or shRNA-empty vector (mock). (**A**–**E**) Total RNA was extracted and *E7* (**A**), *RB1* (**B**), *E2F1* (**C**), *MCM2* (**D**), and *PCNA* (**E**) mRNA was evaluated by real-time RT-PCR at 48 h post-transfection. (means ± SD; *n* = 3) (**F**) Cell cycle analysis by flow cytometry. Representative cytograms of the cell cycle distribution at G1 (red), S (yellow), and G2 (green) phases are shown. * *p* < 0.05, ** *p* < 0.01.

**Figure 4 cancers-13-04613-f004:**
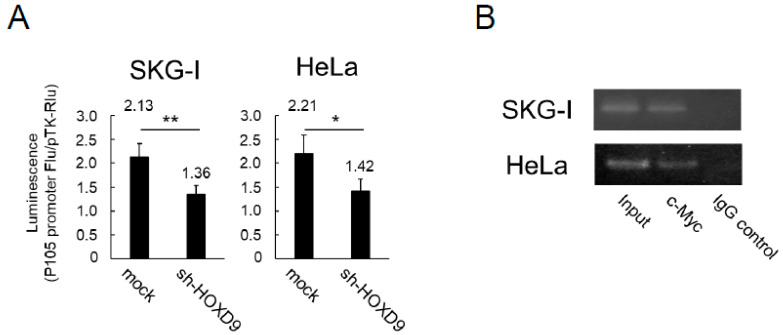
HOXD9 binds directly to the P105 promoter and regulates function. (**A**) Relative luciferase activities of *P105* promoter reporters in vector control (mock) and *HOXD9*-knockdown CC cells (sh-HOXD9). Data are expressed as the mean ± SD of three replicates. (**B**) Binding of HOXD9 to the *P105* promoter was analyzed by a ChIP assay, as described in the Materials and Method section. * *p* < 0.05, ** *p* < 0.01.

**Table 1 cancers-13-04613-t001:** The top 20 activated upstream regulators predicted by IPA.

Upstream Regulator	Molecule Type	Activation z-Score	*p*-Value of Overlap
IL1RN	cytokine	4.298	0.00197
epigallocatechin-gallate	chemical drug	4.192	0.00392
NKX2-3	transcription regulator	4.135	0.000607
Irgm1	other	3.978	0.00673
TP53	transcription regulator	3.830	1.1 × 10^−19^
CDKN1A	kinase	3.754	0.00171
fulvestrant	chemical drug	3.603	0.000958
U0126	Chemical-kinase inhibitor	3.494	0.000703
PTGER4	G-protein coupled receptor	3.342	0.0123
gentamicin	chemical drug	3.219	0.0013
PTEN	phosphatase	3.177	0.000364
RB1	transcription regulator	2.889	0.0024
fenretinide	chemical drug	2.866	0.00309
MEOX2	transcription regulator	2.853	0.00603
RPSA	translation regulator	2.726	0.000899
SP600125	Chemical-kinase inhibitor	2.718	0.00125
IgG	complex	2.705	0.0122
ACKR2	G-protein coupled receptor	2.673	0.00926

## Data Availability

The authors confirm that the data supporting the findings of this study are available within the article and its Appendix A.

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
