# Peer review of "Transcription Factor Homeobox D9 Drives the Malignant Phenotype of HPV18-Positive Cervical Cancer Cells via Binding to the Viral Early Promoter"

_cancers, 2021, doi:10.3390/cancers13184613_

Round 1

Reviewer 1 Report

Hayashi et al investigated the role of the transcription factor HOXD9 in the regulation of the expression of two human papillomavirus (HPV)18 virus early genes, E6 and E7, in two cervical cancer cell lines (HeLa and SKG-I). The authors found that knocking down HOXD9 results in a decreased cellular proliferation and in the downregulation of the viral genes E6 and E7. Downregulation of E6 is associated with a stabilization of p53 and an induction of apoptosis while downregulation of E7 decreases the expression of two E2F target genes, MCM2 and PCNA, and an increase in G1 cells. Microarrays following HOXD9 knockdown also show an increase in expression of genes associated with p53 and RB pathways while the E2F pathways are downregulated. The authors then show in a luciferase assay that HOXD9 knockdown decrease expression of the p105 promoter (promoter of the early HPV genes) and that HOXD9 binds to the p105 promoter in a ChIP assay.

While the manuscript shows interesting results, several important controls are missing to support the authors conclusions.

Comments:

  1. Only one HOXD9 shRNA was used in this study, which raises concerns on the specificity of this shRNA. It would be important to confirm the key results with a second HOXD9 shRNA.
  2. The authors should perform qPCR for the ChIP experiment. Also, the authors used a HOXD9-Myc construct for the ChIP experiment so they will need to perform a ChIP experiment in absence of this construct to show that Myc is not binding to the p105 promoter. Currently, they can’t differentiate between HOXD9 or Myc in their ChIP experiment. Also, it would be important to have a negative primer to see the background level of the ChIP.
  3. Western blot of HOXD9 in control and after shRNA is required to show protein degradation. Decrease of the mRNA is not obligatorily associated with a strong decrease in protein level if the protein is extremely stable.
  4. An important control would have been to knockdown HOXD9 in a non-HPV cell line to control whether the effects the authors observed are mediated through HPV18 E6 and E7 proteins or by something else (typically, if the authors also see a decrease in cellular proliferation and an increase in apoptosis in the non-HPV cell line, it will clearly indicate that the effects they observed in this manuscript is not (only) mediated through E6 and E7).
  5. Figure 1C: the experiment has been performed only once. Biological replicates are required and the authors should add error bars and statistical tests. It would also strengthen the manuscript to have different time points in the cellular proliferation assay.
  6. Figure 2B: Western blot of the E6 and E7 proteins are required.
  7. Most of the qRT-PCR experiments seem to have been performed only once (Figure 2B-C, Figure 3B-E). Biological replicates are necessary and the authors should also add error bars and statistical tests in these figures.
  8. The authors need to provide as a supplementary table the list of genes from their microarray experiments. The authors need to also say how many biological replicates has been performed and in which cell lines. Data should also be uploaded to the GEO database to allow other researchers to confirm the authors claim.
  9. Figures 1A, 2A, and 3A are not cited in the manuscript.
  10. Line 217: It is figure 2D, not 3D.
  11. Lines 217-219: HOXD9 is not involved in the degradation of p53, p53 stability is regulated by E6, which is in turn regulated by HOXD9 based on the authors claim.
  12. Line 226: Knockdown of HOXD9 induces apoptosis by activating the p52 pathway, not by suppressing it as the authors wrote.
  13. Line 230: 3D gene analysis: it is unclear what the authors mean by that.
  14. Line 304: HOXD9 knockdown results in a stabilisation of the p53 protein, not an increase in the expression of p53 protein.

Reviewer 2 Report

Cervical cancer is caused by sexually acquired infection with certain types of HPV. Two HPV types (16 and 18) cause 70% of cervical cancers and pre-cancerous cervical lesions. There is also evidence linking HPV with cancers of the anus, vulva, vagina, penis and oropharynx. The real cause of cervical cancer is not the HPV infection per se. Over-expression of E6 and E7 oncoproteins is a critical and required.

Hayashi et al. have previously demonstrated that homeobox D9 (HOXD9) transcription factor is responsible for the malignancy of HPV16-positive cervical cancer cell lines via binding to the P97 promoter. Now they show that HOXD9 drives the HPV18 early promoter activity to promote proliferation and immortalization of cervical cancer cells.

The claims are properly placed in the context of the previous literature. The experimental data support the claims. The manuscript is written clearly enough that most of it is understandable to non-specialists. The authors have provided adequate proof for their claims, without overselling them. The authors have treated the previous literature fairly. The paper offers enough details of methodology so that the experiments could be reproduced.

Minor revisions

Page 2, line 46, "Cervical cancer (CC) is the third most frequent cancer in women worldwide" => "Cervical cancer is the fourth most common cancer in women"

https://www.who.int/health-topics/cervical-cancer#tab=tab_1

Page 2, line 49-52, "There are approximately 40 types of HPV that infect the female genital organs, 15 types of which, HPV16, 18, 31, 33, 35, 39, 45, 51, 52, 56, 58, 59, 68, 73, and 82, are considered carcinogenic and classified as high-risk types."

There are 20 HPV genotypes that are known to cause cervical cancer  (oncogenic of high-risk types).

Arbyn M, Tommasino M, Depuydt C, Dillner J. Are 20 human papillomavirus types causing cervical cancer? J Pathol. 2014 Dec;234(4):431-5. doi: 10.1002/path.4424. PMID: 25124771.

https://pubmed.ncbi.nlm.nih.gov/25124771/

Round 2

Reviewer 1 Report

The authors have answered my comments.